# Single-shot link discovery for terahertz wireless networks

Yasaman Ghasempour [1], Rabi Shrestha[2], Aaron Charous [2], Edward Knightly[1] & Daniel M. Mittleman [2]✉

Of the many challenges in building a wireless network at terahertz frequencies, link discovery remains one of the most critical and least explored. In a network of mobile receivers using narrow directional beams, how do the nodes rapidly locate each other? This direction information is crucial for beam forming and steering, which are fundamental operations for maintaining link quality. As the carrier frequency increases into the terahertz range, the conventional methods used by existing networks become prohibitively time-consuming, so an alternative strategy is required. Using a leaky-wave antenna with a broadband transmitter, we demonstrate a single-shot approach for link discovery which can be accomplished much more rapidly. Our method relies on measurements of the width of a broad spectrum, and does not require any information about the phase of the received signal. This protocol, which relies on a detailed understanding of the radiation from leaky-wave devices, offers a realistic approach for enabling mobility in directional networks.

[1] Department of Electrical and Computer Engineering, Rice University, 6100 Main St., Houston, TX 77005, USA. [2] School of Engineering, Brown University, 184 Hope St., Providence, RI 02912, USA. ✉email: daniel_mittleman@brown.edu

The development of wireless networks employing frequencies above 100 GHz has become a topic of intense research activity[1–3], offering both challenges[4] and opportunities[5]. In such a network, where communication channels employ narrow directional beams[6], one of the most important protocols for an access point is to determine the direction to each receiver[7–9]. To allow for the possibility that receivers could be mobile, this information must be harvested rapidly and repeatedly, without requiring substantial overhead that would significantly impact data transmission rates. At lower frequencies in the millimeter-wave range, the conventional approach relies on a sequential search, in which the nodes step through all angles until a link is discovered[10]. At higher frequencies, the beams become narrower so more steps are required for this discovery process, rendering this sequential approach impractical[11,12]. An alternative method is suggested by the use of leaky-wave antennas, which impose a coupling between the frequency of emitted radiation and the emission angle.

Leaky-wave devices have been employed in microwave systems for many years, although their investigation in the terahertz range is much more recent[13–16]. In a typical guided wave implementation of such devices, a signal propagating in a waveguide can couple through an aperture to "leak" out into free space, if the guided mode and the free-space mode satisfy a phase-matching condition on their parallel wave vector components. For the simplest architecture, a metal parallel-plate waveguide with empty space between the plates, with the lowest-order transverse electric ($TE_1$) mode propagating in the waveguide, this phase-matching requirement imposes a constraint on the angle of propagation of the radiation emitted from the aperture, for a given frequency $f$:

$$f(\phi) = \frac{f_c}{\sin \phi}, \tag{1}$$

where $f_c$ is the waveguide cutoff frequency, given by $c_0/2b$, and $\phi$ is the propagation angle of the free-space mode relative to the waveguide propagation axis. Here, $b$ is the plate separation and $c_0$ is the vacuum light velocity. This result implies that, if the leaky waveguide (LWG) is excited with a broadband source, then each frequency within this broad spectrum should emerge from the slot aperture at only one unique angle. In other words, a detector looking at the emitted radiation with a small aperture should detect only a very narrow spectrum, whose bandwidth is determined only by averaging Eq. (1) over the detector aperture.

To lowest order, this result provides an adequate description of the behavior of leaky-wave devices[13], and this approach has been the basis of implementations in communications[16] and radar[15,17] demonstrations in the terahertz range. However, this simple description neglects important effects, such as the thickness of the metal plate containing the leaky-wave aperture and the finite length of the emission region[18], which can have a dramatic influence on the spectrum of emitted radiation at a given angle. A more careful analysis suggests a more complex relationship between frequency and angle, such that a range of frequencies can be detected at any given angle.

We note that a variety of different system architectures have previously been considered for realizing efficient link discovery. For example, the use of legacy bands at lower frequencies has been proposed, leveraging existing infrastructure[19–21]. Other architectures exploit assumptions about motion correlations, requiring prior knowledge about client locations[11,22]. None of the aforementioned methods can help determine both client location and rotation.

Here, we show that the broad spectrum emitted from a LWG enables a new method for link discovery for an access point in a local area network, including both the angular location and the rotation angle of the mobile client (i.e., both angle of departure

and angle of arrival). Angle of departure (AoD) information can be obtained from the frequency of the spectral peak of the signal received by the client. Client rotation (angle of arrival, AoA) can be determined from the high-frequency and low-frequency edges of the received spectrum. This information can be harvested rapidly, using a single pulse of broadband emission from the access point, and requires no information about the spectral phase of the received signal.

## Results

**Bandwidth of received signal.** The idea for obtaining directional information is illustrated schematically in Fig. 1. Both the transmitter (e.g., access point) and the mobile receiver (client) are equipped with leaky-wave waveguides. The transmitter excites the $TE_1$ mode of the waveguide with a broadband source[23], whose spectral coverage is broad enough to illuminate the entire relevant angular range, according to Eq. (1). The LWG fills the space with a range of frequencies, in the form of a THz "rainbow". If the client's waveguide is parallel to the transmitter's waveguide, then it is clear that a signal at a particular frequency will couple into the waveguide. However, if the client is rotated, then the two angles do not match. In this case, using the simple analysis of Eq. (1), one would expect that the client would receive no signal, even for a very small rotation away from perfectly parallel. This is why a more sophisticated analysis of the leaky-wave device is necessary; the spectrally broader emission at a specific angle enables a finite range of client rotation without complete loss of signal.

We can understand this broader spectral width in two ways. First, one can treat the leaky-wave slot as a finite-length aperture, which produces a diffraction pattern in the far field. In this case, the angular distribution of the diffracted field (in the plane of the

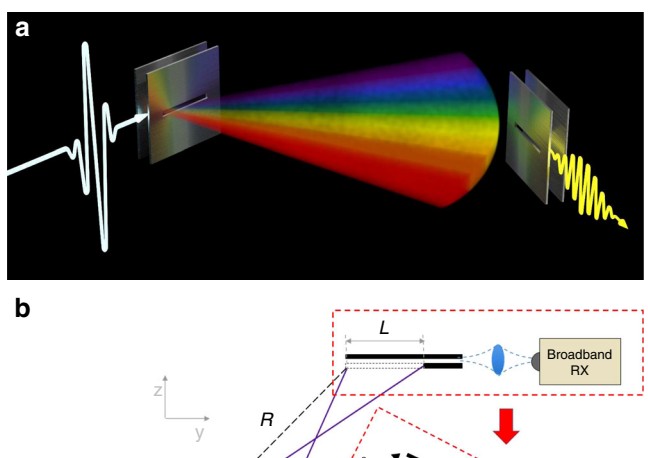

**Fig. 1 Drawings of the experimental arrangement. a** An illustration showing the "THz rainbow" emitted by a leaky waveguide (LWG) after excitation of the waveguide with a broadband input signal. A receiver (RX), also employing a LWG, captures a portion of this broadband signal. The range of the received spectrum can be used to determine, not only the angle at which the receiver is located, but also the rotation of the receiver relative to the transmitter. When a broadband transmitter (TX) is used, the client's position can be obtained in a single shot. **b** A schematic of the experiment, in a plan view, defining various parameters used in the discussion, and illustrating the rotation of the RX waveguide.

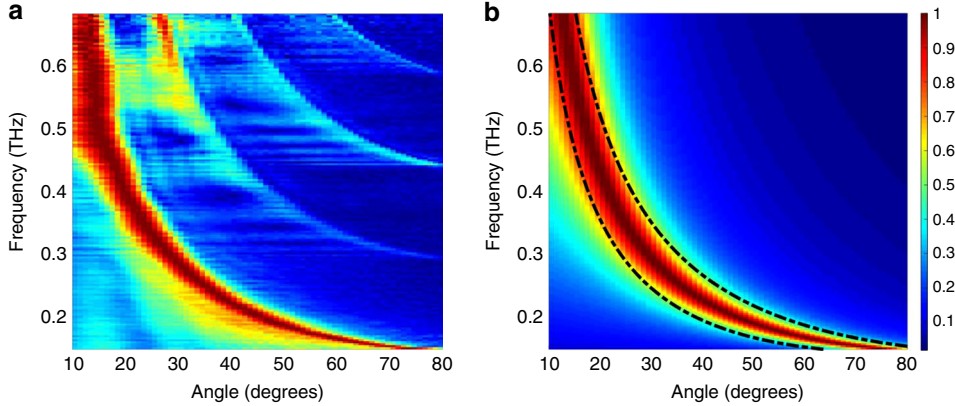

**Fig. 2 The spectrum of emitted radiation vs. emission angle. a** A plot of the spectrum of the radiation emitted by a LWG at emission angle $\phi_0$ (defined in Fig. 1b), after excitation with a broadband input. This is measured using a broadband detector staring directly at the emission point, without a second LWG. Each row of this image has been normalized to unity magnitude, in order to remove the frequency-dependence of the input signal from the THz-TDS transmitter, and emphasize the signals at higher frequency. The prominent arc in the lower left region corresponds to the emission from the dominant TE$_1$ waveguide mode; the weaker arcs in the upper right arise from higher-order TE waveguide modes (TE$_2$, TE$_3$, and TE$_4$), which result from imperfect input coupling to the waveguide. The TE$_1$ mode signal represents about 90% of the total radiated energy. **b** Two different predictions of the measured spectrum-angle relation displayed in **a**. The false color plot corresponds to a diffraction model (Eq. 2) using a value of $\alpha = 0.213\,\text{mm}^{-1}$ for the leakage parameter, while the two dotted black curves represent the results of a ray optics model (Eq. 3), with the numerical parameter values discussed in Supplementary Note 1.

slot) is given by:

$$|E(\phi)| = \text{sinc}[(\beta - i\alpha - k_0\cos\phi)(L/2)], \qquad (2)$$

where $\text{sinc}(x) = \sin(x)/x$, $\beta$ is the wave vector of the TE$_1$ guided wave, $k_0 = \omega/c$, $L$ is the slot length, and $\alpha$ is a parameter which describes the loss of energy in the guided mode due to leakage out of the slot[14,18].

An alternative description relies on ray optics. For a LWG with infinitely thin metal plates, the energy leakage is determined only by phase matching. However, for a plate of finite thickness, the slot itself acts as a waveguide, which presents an impedance boundary between the TE$_1$ fast wave and free space. Rays can reflect from this boundary, and remain in the waveguide for a longer propagation distance before leaking out. As illustrated in Fig. 1b, this results in a larger effective length for the emission region. From geometrical considerations (See Supplementary Note 1), we derive the minimum and maximum angles at which a light ray could be received, as:

$$\theta_{\min} = \tan^{-1}\left(\frac{k_z R}{k_y R + k_0 L}\right) \quad \theta_{\max} = \tan^{-1}\left(\frac{k_z R}{k_y R - k_0 L}\right), \qquad (3)$$

Here, $R$ and $L$ are defined in Fig. 1b, and $k_y = \sqrt{k_0^2 - k_z^2}$ and $k_z = \pi/b$ are the $y$ and $z$ components of the free-space wave vector, respectively. We assume an effective slot length $L$ which is identical for both transmitter and receiver. We note that this ray optics approach makes sense only in the limit where the rate of emission is large, such that the loss parameter $\alpha$ satisfies $\alpha L > 1$. This limit is somewhat unusual for most previous considerations of leaky-wave devices[18].

Both the diffraction formalism and the ray optics picture can be used to predict the spectral bandwidth of radiation emitted at any given angle from the leaky-wave slot, assuming that the waveguide is excited with a broadband input. Figure 2 shows their agreement with each other, and with results measured using the test-bed system described below. Since our approach to client location and rotation sensing described below relies only on determining the peak and upper and lower limits of the received

spectrum, we rely on the ray optics approach (Eq. 3) for subsequent discussion, in the interest of computational simplicity.

**Angle of departure.** Based on these results, we develop an approach for locating a mobile receiver in the far field of the transmitter (access point). This receiver (client) can detect only a portion of the THz rainbow. This subset of the transmitted spectrum contains information about the line-of-sight angle of the client relative to the access point. We focus on the spectral peak of the received signal and translate it to the corresponding angle using Eq. 1. Note that this approach does not require any prior knowledge other than the geometry of the LWG (i.e., the plate separation). Further, it requires only power measurements at the receiver and not phase information. This dramatically simplifies the THz node architecture, eliminating the need to keep tight synchronization between the transmitter and receiver, and is robust to small-scale channel variation.

To explore the effectiveness of this protocol, we have built a scale-model test bed (illustrated in Fig. 1b) based on a conventional terahertz time-domain spectrometer to provide access to a continuous broadband spectrum ranging from 100 GHz to over 1 THz, in the form of a short pulse[2,24]. We note that this is a very low-power source[25], so the transmission range of the test bed is limited to a few tens of cm. Clearly, with the higher power available from broadband emitters based on integrated devices[26–28], a larger range could be achieved. Nevertheless, this test-bed setup is sufficient for demonstrating the efficacy of the new link discovery protocol described above. For our LWG, we use aluminum plates with a spacing of $b = 1.04\,\text{mm}$. We excite the TE$_1$ mode of this waveguide by quasi-optic coupling from a focused Gaussian beam. For detection, a broadband receiver is located in the far field of the LWG output, staring directly at the emission point. More details on the experimental procedures can be found in the "Methods" section, below.

We measure the received spectrum for many different locations of the receiver, and extract an estimate of its angular location from these spectra. The results are summarized in Fig. 3, demonstrating an average estimation error of only a few degrees. The error increases somewhat for larger values of $\phi$, because of the reduced variation of frequency with angle (see Fig. 2a), and

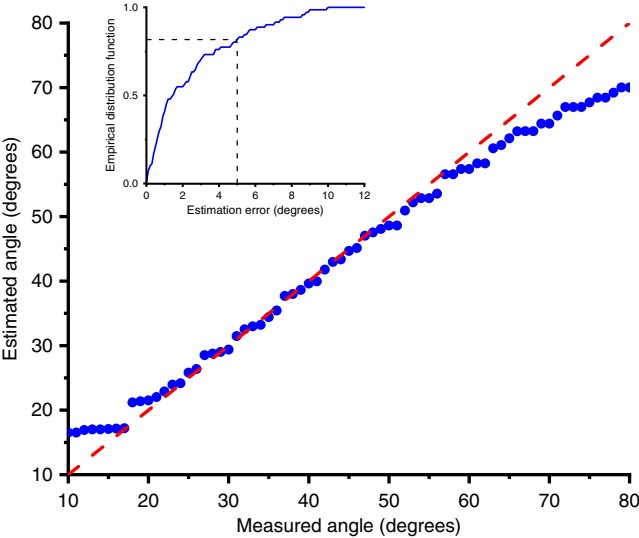

**Fig. 3 The accuracy of single-shot angle-of-arrival extraction.** This plot compares the angle of the client extracted from the peak frequency of the measured spectrum against the actual angle (dashed red line), which is obtained from physical measurement of the setup. Here, unlike in Fig. 2, the spectra are obtained with a LWG at both the transmitter and receiver. The two waveguides are oriented parallel to each other, such that $\theta_{rot} = 0$. The inset shows the empirical distribution function. The dashed black lines indicate that the estimation error is less than 5° in more than 80% of measurement instances. Source data are provided as a Source Data file.

also because of the finite spectral resolution of our measurement system (about 3 GHz).

**Angle of arrival**. An even more exciting prospect is to use the spectral width of the received spectrum for estimating the rotation of the client[11,22,29]. To explore this idea, we consider a client located at a particular angle $\phi_0$ relative to the access point. In this case, we modify the detection by adding a LWG at the receiver in addition to the one at the transmitter (i.e., as illustrated in Fig. 1). If these two waveguides are parallel, the AoD from the transmitter is equal to the AoA at the receiver. In this case, as discussed above, the low- and high-frequency edges of the spectrum are determined by $\phi_0$. However, if the client is rotated by an angle $\theta_{rot}$, the AoA shifts by $\pm\theta_{rot}$ (+ for clockwise CW rotation, − for counterclockwise CCW rotation). Hence, for CW (CCW) rotation, the upper (lower) edge of the received spectrum shifts to lower (higher) frequencies. We find the magnitudes of these shifts are:

$$\left.\frac{\partial f_{max}(\theta)}{\partial \theta}\right|_{\theta=\phi_0} = \frac{f_{max} - f_{max,\theta_{rot}=0}}{\theta_{rot}} \quad \text{CW case}$$
$$\left.\frac{\partial f_{min}(\theta)}{\partial \theta}\right|_{\theta=\phi_0} = \frac{f_{min} - f_{min,\theta_{rot}=0}}{\theta_{rot}} \quad \text{CCW case} \quad , \quad (4)$$

(see Supplementary Note 2 for derivation). Thus, we can extract the rotation angle from measurements of the high- and low-frequency edges of the spectrum. Comparisons between our test-bed measurements and the predictions of Eq. 4 are shown in Fig. 4a, b. Obviously, if the rotation angle $\theta_{rot}$ is too large, then no spectral information is received, and the rotation angle cannot be determined. However, for a surprisingly large range of angles (which depends on $\phi_0$, as shown in Fig. 4c), rotation can be accurately tracked using a single-shot measurement. Once again, the average estimation error is in the range of just a few degrees.

## Discussion
We have developed an approach for locating a mobile receiver in a highly directional network which does not rely on a sequential trial and error method, but rather can be accomplished in a single shot using broadband sources and receivers. This method enables the sensing of both the angular location and angular rotation of a client, both of which are necessary for two-way communication using directional beam forming[12,22]. Recognizing that coherent detectors are significantly more challenging to build and operate, we anticipate that future system designers will prefer to rely on approaches which do not require detection of the phase, but rather only of the amplitude. Therefore, although the experiments described here are sensitive to the phase of the terahertz field, we emphasize that our method relies only on knowledge of the power spectrum of the measured signals, and not their spectral phase.

The protocol described here has a number of interesting advantages, it is scalable to multiple clients within the angular view of a single access point, since each client would employ a different portion of the THz rainbow. Although we excite the transmitter's LWG in a single-sided configuration, it would be possible to excite the device from both sides, thus doubling the accessible angular range, simply by adding a second independent transmitter on the opposite side of the LWG. Further, a more complicated LWG slot geometry could be employed to cover a larger range of angles, or out-of-plane paths (i.e., the third dimension). This general approach also has the potential to be able to identify non-line-of-sight paths. At least in the case of a sparse scattering environment, a small number of signals arising from specular reflections in the environment would each produce a signal with a unique spectral signature, as they each arrive at the receiver from a different angle. Finally, we emphasize that, because the proposed protocol only requires a single burst from a broadband source (rather than a series of sequential signals), it is compatible with the low-latency requirements that will be mandatory in 5G and beyond. It can therefore enable real-time client tracking in a directional network.

## Methods
**Experimental procedure**. The measurements described in this work were performed using a scale model of a transmit–receive system as illustrated in Fig. 1. As noted above, the size of this test bed (i.e., the range of transmission) is limited by the low power-per-unit-bandwidth of the transmitter—a typical terahertz time-domain spectrometer (THz-TDS) with fiber-coupled photoconductive antennas produces about 10 μW of power, integrated over the entire spectral range from 100 GHz to 2 THz[23]. A rough back-of-the-envelope estimate indicates that this corresponds to less than −130 dBm/Hz at the peak of the spectrum. The broad bandwidth produced by this spectrometer is ideal for characterizing broadband components and subsystems. However, a real system implementation would need to employ transmitters that operate over a narrower (but still very broad) spectral range, but with higher output power, and which are more amenable to integration in a compact package[24–26]. The time-domain system used here is nevertheless valuable for demonstrating the feasibility of the single-shot measurement concept. As with all THz-TDS systems, the data consists of a time-domain waveform which is proportional to the THz electric field, and which can therefore be numerically processed via Fourier transform to obtain both spectrum and spectral phase, over the entire accessible bandwidth. In the results reported here, we do not make use of any of the measured spectral phase information.

In our test-bed experimental realization, we employ bulk metals for both of the leaky-wave waveguide structures. We note that a low-loss low-dielectric slab with metallic coatings on both sides would work equally well, and would provide a smaller form factor suitable for both fixed and mobile devices. Our plate spacing ($b = 1.04$ mm) corresponds to a TE₁ cutoff frequency of ~144 GHz. The quasi-optic input coupling, if optimized, can theoretically result in ~99% coupling efficiency to the lowest-order TE₁ waveguide mode[23]. In our case, the presence of higher-order TE modes is confirmed by the higher arcs in Fig. 2a. By integrating subsets of this two-dimensional data set, we estimate that about 90% of the energy emerging from the slot comes from the TE₁ mode, with the remaining 10% distributed among higher-order modes.

For detection, we use the receiver arm of the THz-TDS system; for the data of Fig. 2a, this receiver is staring directly at the slot in the TX LWG, but for the subsequent data, this receiver is coupled to the output of a second LWG, which is used to collect the signal as illustrated in Fig. 1a. This second waveguide is constructed with the same slot size and same plate spacing as the TX LWG.

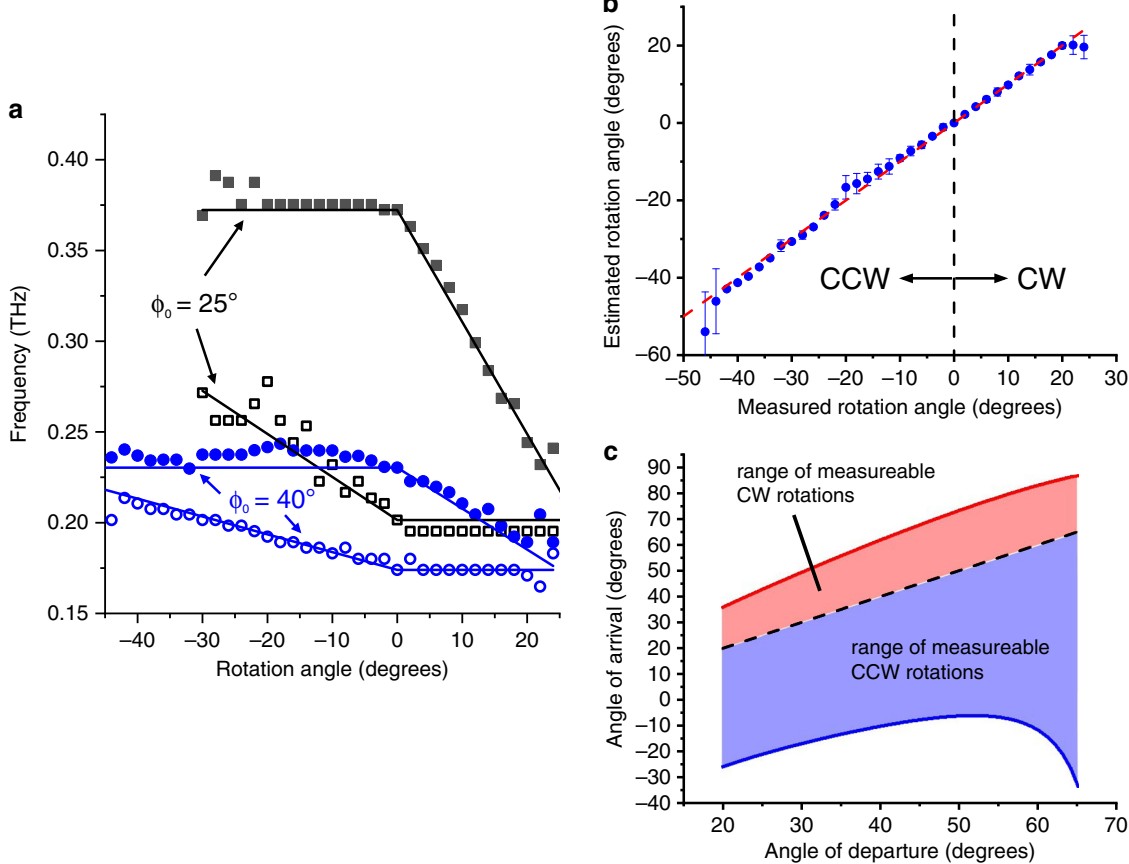

**Fig. 4 Characterization of client rotation.** For a given angle at which the receiver (client) is located $\phi_0$, we extract an estimate of the client's rotation angle $\theta_{rot}$, from the high-frequency ($f_{max}$) and low-frequency ($f_{min}$) edges of the measured spectra. **a** The extracted values of $f_{max}$ and $f_{min}$ as a function of client rotation angle, for two different values of $\phi_0$. The solid lines represent the predicted values based on ray optics (Eq. 4). These results indicate the typical level of agreement between measurement and prediction, over the range of values of $\theta_{rot}$ where a nonzero spectral width is predicted by the theory. **b** By compiling all measurements at a given rotation angle (at each $\phi_0$), we extract the measurement uncertainty in the rotation angle as a function of the degree of rotation. The data points represent the average values of the extracted rotation angles at many values of $\phi_0$; the error bars indicate the standard deviations of these averages. The uncertainty increases somewhat for larger rotations, since a smaller signal is measured for larger rotations. Nevertheless, over the range of accessible rotation angles, an average estimation error of less than 2° is obtained. **c** From the ray optics theory, we can predict the maximum values of $\theta_{rot}$ which can be sensed for any given client location $\phi_0$. This is not symmetric with respect to the direction of rotation (CW vs. CCW), because of the asymmetry of the emission configuration (i.e., $0 < \phi_0 < 90°$). However, as noted in the text, a second transmitter, positioned on the opposite end of the LWG, would produce a symmetric emission pattern at angles $90° < \phi_0 < 180°$, which would symmetrize the rotation sensing. Source data are provided as a Source Data file.

Because of the fiber-coupling of the antennas, we are able to easily reposition the receiver/LWG subsystem to characterize the emitted signal at any desired angle. Our measurements span a range of roughly 70° in $\phi_0$, as the emission efficiency from a LWG drops dramatically when the angle is outside of the range 10°–80°. Our measurement system provides a spectral resolution of about 3 GHz, limited by the length of a scanning delay line. This is more than adequate to resolve the measured spectra displayed in Fig. 2a, even at large values of $\phi$ where the spectrum is at its narrowest.

**Atmospheric effects.** In our discussion, we have neglected the effects of attenuation of signals due to atmospheric water vapor. As is well known, water vapor gives rise to several narrow absorption lines in the THz range. It is generally envisioned that systems will be designed in the windows between these narrow lines, avoiding their influence. However, for a propagation range of, e.g., 100 m (as one might typically envision for an indoor local area network), even the strongest water absorption lines at frequencies below 500 GHz add a relatively small contribution to the overall propagation loss (atmospheric attenuation + free-space path loss). For the transmission ranges accessed in our measurements (<1 m), these absorption lines are negligible.

## Data availability
The data that support the findings of this study are available from the corresponding author upon reasonable request. Source data for Figs. 3 and 4 are provided with the paper.

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

## Acknowledgements

This work was supported in part by Cisco, Intel, and by the US National Science Foundation.

## Author contributions

All of the authors contributed to the conception and design of these experiments. Y.G. and R.S. built the measurement setup and acquired the data. A.C. and Y.G. developed the ray optics model. All authors contributed to the discussion and writing of the paper.

## Competing interests

The authors declare no competing interests.

## Additional information

**Peer review information** *Nature Communications* thanks Bile Peng and the other, anonymous, reviewer(s), for their contribution to the peer review of this manuscript. Peer reviewer reports are available.

