## [Peer Review File · Nature Communications]

Reviewers' comments:

Reviewer #1 (Remarks to the Author):

This paper presents an interesting concept on link discovery, angle and rotation estimation. In general, the problem addressed is important and has not yet been exhaustively studied. The proposed method makes sense. I have the following comments to further improve the quality of the paper:

- Authors simplify the considered problem to the ray optics approach (page 4). However, if the propagation path is NLoS (e.g., via reflection or scattering), the geometric relationship shown in Fig. 1 does not hold any more. In a more complicated case, what if there are several multipath components (MPCs) with similar channel gains? Would the proposed method still work? (if not, can it be combined with eigenvalue based algorithms such as MUSIC and ESPRIT because the signals with different frequencies can be considered uncorrelated?)

- I understand there is a page limit, but it would better to briefly describe the experiment in the paper rather than putting all info in the supplementary materials, such that readers can understand it better.

- In Fig. 3, I understand the reason why the error increases for large degrees, but is there an explanation why errors are also large for angle smaller than 15 degree? Is it because the red spectrum is broader for small angles?

- At the end of the paper, the authors consider the rotation estimation but limit the rotation in the two-dimensional space. In the three-dimensional space, the rotation is described by three angles: yaw, pitch and roll rather than one. Please refer to Section III-B in [19] (your reference) for the calculation, or clarify that your method is limited to 2D.

Bile Peng

Reviewer #2 (Remarks to the Author):

A. Summary of the key results

The authors report on a method for link and cell discovery at THz carrier frequencies in the range 0.1 THz to 0.7 THz using a single-shot approach. The discovery signal is transmitted and received with leaky-wave antennas. Angle of departure and angle of arrival can be determined. The authors claim that their method "offers the first realistic protocol for enabling mobility in directional networks".

Basically, the paper is highly interesting and deserves publication, if certain very important and crucial data are supplemented, if these data prove the feasibility of the method, and if the authors' proposal is put in perspective by discussing competing technologies. One highly interesting approach is to have a high-speed user plane with THz carriers, and to discover the user location by an initial search employing a legacy control plane with only slow data transmission on GHz carriers [Filippini 2018, Fast Cell Discovery in mm-Wave 5G Networks with Context Information, IEEE TRANSACTIONS ON MOBILE COMPUTING, VOL. 17, NO. 7, JULY 2018, 1538-1552, doi:10.1109/TMC.2017.2772881]. Another method is described in [11], a reference which the authors just listed thrice.

B. Originality and significance

A leaky wave antenna radiates in different directions, depending on frequency. The authors generate a THz pulse, the temporal width of which would be about 1.7 ps according to the spectral range specified in Fig. 2. At the user terminal, they measure the change of the high-frequency and the low-frequency edges of the received spectrum when rotating the receiver, and infer both, the angle of departure and the angle of arrival. In their experiments they put a receiver at a distance of 15 cm away from the transmitter. Transmitter and receiver antennas consist of a metallic parallel-plate waveguide (spacing 1.04 mm, TE1 cutoff frequency 144 GHz) with a radiating (receiving) slot of 3 cm length. A Fourier transform spectrometer with a resolution of 3 GHz records the spectra.

The emission efficiency from the leaky wave antenna drops outside the angle range 10° to 80° , but in this 70° range the authors show that in 80 % of the cases of equally distributed angles the angle estimation error was smaller than 5° , while in the rest of the cases it could be as large as 10° . This has to be seen on the background of highly directed beams from steerable arrays, where moderate antenna gains compared to an isotropic radiator of $G_i = (15, 21, 27, 33, 39)$ dBi correspond to beam widths of $\theta_i = \{20^\circ, 10^\circ, 5^\circ, 2.5^\circ, 1.25^\circ\}$.

Atmospheric loss was not regarded by the authors, but could be of importance, because at some frequencies in their pulse spectral range the loss is as high as 200 dB/(10 m) (e.g., at 0.56 THz).

The approach of the authors is original and could be of practical significance - but this has to be proved yet.

C. Overall evaluation and recommendations

- Essential data are missing: Pulse power or energy, pulse duration, centre frequency, radiated power, radiated beam shape, received power at relevant distances (150 m or 15 m or at least 1.5 m instead of 15 cm I would regard as relevant)
- The claim of the authors is that the "single-shot approach" is more rapid than sequentially stepping through different beam steering directions using CW radiation. This has to be quantified: What would be the beam stepping time (the steering can be done electronically!), and how long is the acquisition and processing time in the present case, until a "single shot" is spectrally evaluated? How would the authors substantiate the claim of "low-latency client tracking, for the first time"?
- How would the angle accuracy depend on the signal-to-noise power ratio at the receiver? I imagine that the single shot has to be periodically repeated for improving the signal-to-noise power ratio sufficiently, and possibly, if really a single shot is evaluated, it need to be (digitally?) stored for performing the Fourier transform of the ACF.
- What THz power would be needed to discover a base station or a user terminal at distances up to 150 m? What would be the energy requirement, would it not be excessive for a battery-operated user terminal? What is the spectral density at the receiver side?
- For the base station, energy considerations are less important, however, the mobile user needs again to have both, a pulsed and a modulated CW THz source. Possibly, the method [Filippini] which could rely on the conventional mobile network for transmitting the localisation of the subscriber or the base station would be more efficient. The present single-shot approach should be put in perspective and compared with the methods [Filippini] and [11] as mentioned at the end of Section A.
- In this context, I strongly contradict the anticipation of the authors that future THz systems will rely on amplitude or intensity modulation only. Good arguments should be given for such an

apodictic statement. I feel that it would be either a waste in spectral resources, or it would require electronic circuits with significantly larger bandwidths. It would be the same idea as the past notion that with the advent of optical amplifiers optical intensity modulation would suffice for the next umpty years. The tendency today is to use higher-order modulation formats like 4QAM or 16QAM. In mobile communications, 4G LTE uses QAM with OFDM, and for 5G this is a still contender.

D. Further comments in sequential order

-
- What is the authors' definition of a sinc? There are two definitions in use, $\text{sinc}(x) = \sin(x)/x$ or $\text{sinc}(x) = \sin(\pi x)/(\pi x)$
 - The abbreviation PPWG is defined, the abbreviation LWG is not (besides in a caption), and both are used interchangeably.
 - A "highly directional network" and a 5° direction error at a distance of 15 cm (a G_i of only 27 dB corresponds to a beam angle of 5°) seems to be in contradiction.
 - Fig. 2 is called a "false color plot". A standard definition is: "A false-color image is an image that depicts an object in colors that differ from those a photograph (a true-color image) would show." If in a grayscale SEM different structures (silicon, metal etc.) are coloured differently, then it is indeed a false colour image. What the authors have done is a colour-coded 3D plot, where the spectral intensity (unfortunately not indicated at the colour-coding scale) is plotted as a function of frequency and angle (again it would have been nice to see which angle is meant - this could have been indicated in Fig. 1b).
 - In the caption of Fig. 2 it is mentioned that "each row of this false-color image has been normalized to unity magnitude" - I do not understand. Even if I benevolently read "the maximum intensity for each line of constant frequency has been normalized to one" I do not see the justification: The intensity for larger angles will certainly decrease, what harm would it do? And if the scale would suffer, then a logarithmic intensity plot could help.
 - The "conventional terahertz time-domain spectrometer" should be specified. Is it a Toptica instrument? Which model?
 - A "scanning delay line" is mentioned for measuring the ACF for this Fourier spectrometer. How long does the scan take, and what is the electronic processing time? How would it be done for the "rapid link discovery"?

Summary:

In its present state, I cannot recommend publication of the submission. However, the approach is interesting enough that it could be worth reading for a wider audience - if the missing data are supplied, if the approach is put properly in perspective, and if all the claims are substantiated, according to the recommendations listed in points A. to D.

I recommend a major revision.

If space becomes a problem: The (admittedly nice) colourful Fig. 1a could be replaced by a more meaningful schematic setup than is presently displayed in panel Fig. 1b.

Response to referee comments

Reviewer #1 (Remarks to the Author):

This paper presents an interesting concept on link discovery, angle and rotation estimation. In general, the problem addressed is important and has not yet been exhaustively studied. The proposed method makes sense. I have the following comments to further improve the quality of the paper:

We are grateful to the referee for noting the significance and novelty of our work.

- Authors simplify the considered problem to the ray optics approach (page 4). However, if the propagation path is NLoS (e.g., via reflection or scattering), the geometric relationship shown in Fig. 1 does not hold any more. In a more complicated case, what if there are several multipath components (MPCs) with similar channel gains? Would the proposed method still work? (if not, can it be combined with eigenvalue based algorithms such as MUSIC and ESPRIT because the signals with different frequencies can be considered uncorrelated?)

The referee has raised an important question. The possibility of NLOS paths would indeed complicate matters. We offer two points in response. First, unlike at lower frequencies, such paths are sparse – there is no such thing as a ‘rich scattering environment’ at THz frequencies. Therefore, their impact on systems can be, to first order, neglected. However, they cannot and should not be ignored completely, because, for example, they could play a very important role in maintaining links even in the case of transient blockage of the line-of-sight path. We have indeed begun investigating this question experimentally; however, those results are clearly beyond the scope of this first report, which focuses on the most significant and prominently important situation (i.e., LOS). We expect to be writing a subsequent manuscript on the issue of NLOS paths, in the near future. For clarity, **we have added a comment to the concluding paragraph of the manuscript in which we mention the possibility of detecting NLOS paths.**

- I understand there is a page limit, but it would better to briefly describe the experiment in the paper rather than putting all info in the supplementary materials, such that readers can understand it better.

The experimental setup is already illustrated in Fig. 1b. In addition, **we have added a brief discussion of some of the experimental details on pages 5 and 6 of the manuscript.**

- In Fig. 3, I understand the reason why the error increases for large degrees, but is there an explanation why errors are also large for angle smaller than 15 degree? Is it because the red spectrum is broader for small angles?

At angles below about 15 degrees, the emission efficiency of the LWG decreases significantly, leading to a lower signal-to-noise. It may be possible to design a more complicated slot geometry in order to offset this effect; however, in this first report, we use a simple linear slot of constant width, for the purposes of validating the feasibility of the idea. A more complicated slot geometry is a topic of future research, and so a discussion of this point is beyond the scope of the present work.

- At the end of the paper, the authors consider the rotation estimation but limit the rotation in the two dimensional space. In the three-dimensional space, the rotation is described by three angles: yaw, pitch and roll rather than one. Please refer to Section III-B in [19] (your reference) for the calculation, or clarify that your method is limited to 2D.

Indeed, in our experiments, the rotation is limited to 2D. However, two orthogonal LWGs would permit full 3D coverage. Demonstrating that experimentally would be beyond the scope of this first report. However, it is worth a mention. So, **we have added a comment in the conclusion paragraph to note this point.**

Reviewer #2 (Remarks to the Author):

A. Summary of the key results

The authors report on a method for link and cell discovery at THz carrier frequencies in the range 0.1 THz to 0.7 THz using a single-shot approach. The discovery signal is transmitted and received with leaky-wave antennas. Angle of departure and angle of arrival can be determined. The authors claim that their method "offers the first realistic protocol for enabling mobility in directional networks". Basically, the paper is highly interesting and deserves publication, if certain very important and crucial data are supplemented, if these data prove the feasibility of the method, and if the authors' proposal is put in perspective by discussing competing technologies.

We thank the referee for noting the fact that our paper is interesting, and deserves publication. We hope that the modifications to the manuscript, and the comments provided here, will satisfy the concerns expressed by the referee.

One highly interesting approach is to have a high-speed user plane with THz carriers, and to discover the user location by an initial search employing a legacy control plane with only slow data transmission on GHz carriers [Filippini 2018, Fast Cell Discovery in mm-Wave 5G Networks with Context Information, IEEE TRANSACTIONS ON MOBILE COMPUTING, VOL. 17, NO. 7, JULY 2018, 1538-1552, doi:10.1109/TMC.2017.2772881]. Another method is described in [11], a reference which the authors just listed thrice.

We are aware of methods that use legacy bands, as this has been considered by several different authors in the past (including one of us, in 2015). Those methods are quite different from what we have discussed here, enough so that a fair comparison is very challenging. This is particularly true in view of the fact that there is no terahertz communication system or test bed that can be used to make such a comparison. As discussed further below, our intent is to identify a new protocol and show that it works, not to build a system which uses it.

We are a bit puzzled by the comment about reference [11] being 'just listed thrice.' Does the referee mean that we referenced it three times, and should have referenced it more? We do not understand what is the point of that. Or, does the referee mean that the paper by Filippini et al.

referenced it three times, and we should look carefully at that reference? It is unclear.

On the other hand, we do agree that we did not provide a sufficient discussion of previous attempts to develop protocols for identifying the location of a mobile client. This includes the reference by Filippini et al. noted by the referee, along with some other examples. To rectify this situation, **we have added several sentences at the top of page 3, with a few additional references (including that one).** We note that no previous work has ever proposed a method for simultaneously determining both client location and client rotation, as our method does.

B. Originality and significance

A leaky wave antenna radiates in different directions, depending on frequency. The authors generate a THz pulse, the temporal width of which would be about 1.7 ps according to the spectral range specified in Fig. 2. At the user terminal, they measure the change of the high-frequency and the low-frequency edges of the received spectrum when rotating the receiver, and infer both, the angle of departure and the angle of arrival. In their experiments they put a receiver at a distance of 15 cm away from the transmitter. Transmitter and receiver antennas consist of a metallic parallel-plate waveguide (spacing 1.04 mm, TE1 cutoff frequency 144 GHz) with a radiating (receiving) slot of 3 cm length. A Fourier transform spectrometer with a resolution of 3 GHz records the spectra.

The emission efficiency from the leaky wave antenna drops outside the angle range 10° to 80° , but in this 70° range the authors show that in 80 % of the cases of equally distributed angles the angle estimation error was smaller than 5° , while in the rest of the cases it could be as large as 10° . This has to be seen on the background of highly directed beams from steerable arrays, where moderate antenna gains compared to an isotropic radiator of $G_i = (15, 21, 27, 33, 39)$ dBi correspond to beam widths of $\theta_i = \{20^\circ, 10^\circ, 5^\circ, 2.5^\circ, 1.25^\circ\}$.

We note that the comparison mentioned here in the referee's comments is not so valid. In the case of a high-gain antenna mentioned by the referee, that is relevant for a narrowband source. In our case, the source is ultra-broad-band, and the emission pattern depends sensitively on frequency. A fairer comparison might be to compare to the angular width of just one frequency in our 'THz rainbow'. This is a difficult comparison to make experimentally, because 'just one frequency' is impossible to measure when the spectrometer has a resolution of ~ 3 GHz (as well as a finite angular aperture). In any event, it is not clear that the comparison would be meaningful. The important issue is the accuracy with which we can extract the angular location of the client, as compared to the angular width of the *full spectrum* (not of a single frequency) received by that client. Our measurements amply demonstrate that the client receives a spectrum which is sensitive to rotation angle for both clockwise and counter-clockwise rotations. The angle estimation error is dominated by the signal-to-noise limitations of our measurement apparatus, not by the angular width of any spectral component's beam pattern.

Atmospheric loss was not regarded by the authors, but could be of importance, because at some frequencies in their pulse spectral range the loss is as high as 200 dB/(10 m) (e.g., at 0.56 THz).

The effects of water vapor absorption are very well known in this spectral range; obviously, one would never design a long-range communication system at a frequency near 557 GHz. Indeed, there are already *many* publications on the topic of the effects of atmospheric water vapor. We feel that this issue does not need to be rehashed yet again in our manuscript.

To briefly address the issue: atmospheric loss is not considered in our work for a very good reason: it is largely insignificant (compared to the free-space path loss) for a propagation range of up to several hundred meters, for most frequencies that lie below the 0.557 THz water vapor line. This fact is illustrated in the figure shown here. This figure shows the computed atmospheric attenuation (in dB/km) as well as the free-space path loss (FSPL) for three different propagation distances. Note in particular the green curve which shows the sum of these two loss mechanisms for a propagation range of 100 m. It is obvious that the water vapor lines contribute almost nothing (a few dB, except near the 383GHz line where the effect is more like 20 dB in 100 m) in comparison with the FSPL (which is well over 100 dB). This is true for frequencies below 500 GHz.

Our spectrometer continuously spans the range from about 150 GHz to over 1 THz, so our measurements cover a very large spectral range. However, of course we are most concerned with the frequencies that will be used for communications, which means the range below 500 GHz. Also, THz communication systems typically contemplate LANs with less than a few hundred meters of range, where water vapor is not so much of an issue (unless you are sitting right on top of one of the narrow absorption lines, which would be a very foolish choice). Most certainly, on the short ranges accessed in our experiments, water vapor is entirely negligible below 557 GHz. As a result, the effects of the atmosphere are, to lowest order, unimportant in our considerations.

As noted, this issue has really been dealt with already with great thoroughness in the literature. However, to ensure that this point is not ignored, **we have added a brief discussion to the Supplementary Materials concerning the issue of atmospheric losses.**

The approach of the authors is original and could be of practical significance - but this has to be proved yet.

We respectfully disagree. We feel that our work proves the feasibility of our new protocol. We hope that our comments here provide a convincing response to the referee's concerns.

C. Overall evaluation and recommendations

- Essential data are missing: Pulse power or energy, pulse duration, centre frequency, radiated power, radiated beam shape, received power at relevant distances (150 m or 15 m or at least 1.5 m instead of 15 cm I would regard as relevant)

The Supplementary Materials already contain the statement that we have used a conventional

THz time-domain spectrometer. Since this technology has been around for over 30 years, we did not think it was necessary to repeat the performance specifications of these well-known and widely used instruments. However, for completeness, **we have inserted a few sentences, and some new references on pages 5 and 6 of the manuscript, as well as some additional information in the Supplementary Materials.**

- The claim of the authors is that the "single-shot approach" is more rapid than sequentially stepping through different beam steering directions using CW radiation. This has to be quantified: What would be the beam stepping time (the steering can be done electronically!), and how long is the acquisition and processing time in the present case, until a "single shot" is spectrally evaluated? How would the authors substantiate the claim of "low-latency client tracking, for the first time"?

The referee seems to believe that we expect that a THz time-domain spectrometer would be used as the source for the envisioned communication system, in some future implementation. As explicitly stated in the Supplementary Materials, *we do not expect that at all*. Therefore, the speed of our THz-TDS system is irrelevant.

The relevant comparison is *not* to how fast we can make measurements with the system that we are using. Rather, the relevant comparison is between an integrated broadband CMOS source (emitting a single burst of radiation) and an integrated CMOS phased array source (emitting a string of signals sequentially in multiple directions, one after the next). To us, it is self-evident that a procedure which relies on just one signal is faster than a procedure which relies on a sequence of signals which could number in the thousands.

We have added several references on the topic of silicon-based devices which can produce the higher power and broadband signals that would be needed to implement our protocol in an integrated component. The point here is simply to validate that such silicon-based components can indeed be made. To be clear, these are all very recent laboratory-scale demonstrations – the devices are not available commercially (although, we expect that they will be, before the day comes when THz communication systems need to be fielded).

As stated in our manuscript, our work demonstrates the feasibility of a completely new approach to simultaneous AoA and AoD detection. *We do not claim* that we have built a communication system which can operate using this approach with low latency; we *only claim* that we have demonstrated the feasibility of the idea, which is compatible with low-latency requirements because it can be accomplished with a single pulse (i.e., it does not require a sequence of pulses). In view of this, the questions posed by the referee are obviously well beyond the scope of this first report.

- How would the angle accuracy depend on the signal-to-noise power ratio at the receiver? I imagine that the single shot has to be periodically repeated for improving the signal-to-noise power ratio sufficiently, and possibly, if really a single shot is evaluated, it need to be (digitally?) stored for performing the Fourier transform of the ACF.

These are good questions, which, once again, are well beyond the scope of our initial report.

They depend on the details of the CMOS sources that would eventually be used to implement the communication system. We do not have such sources. The questions posed by the referee miss the point, which is that we have demonstrated an entirely new procedure for harvesting necessary information on the location and rotation of a client in a mobile network. This is a first demonstration of feasibility, not a completed system.

In our opinion, it makes no sense to evaluate system performance using a THz source which differs in very fundamental ways from the source that will eventually be used to construct commercially viable systems. What we *can* do, at this stage, is to invent the protocols that this future system will eventually use. We can see if they work; and, we can see if the protocol itself contains anything that would impede its use in those future systems. This is a completely distinct question from those involving source performance. It is inappropriate to entangle those two questions in the very first paper about the protocol.

- What THz power would be needed to discover a base station or a user terminal at distances up to 150 m? What would be the energy requirement, would it not be excessive for a battery-operated user terminal? What is the spectral density at the receiver side?

As stated explicitly in the Supplementary Materials, our measurements have been performed using a scale-model approach, with a *very* low-power source (a back-of-the-envelope estimate gives a value of -135 dBm/Hz at the maximum of our spectrum). Obviously, with higher power, one would be able to make the idea work at greater range. For a first demonstration of feasibility based on such a scale model approach, it hardly seems necessary to perform link budget calculations at 150 m range. That question, and the others posed by the referee, have no bearing on whether or not the idea actually works as a method for extracting AoA and AoD information. To clarify any confusion about the use of a scale-model test bed, **we have moved some of that discussion from the Supplementary Materials to the main text (pages 5-6), where it is more prominently featured.**

- For the base station, energy considerations are less important, however, the mobile user needs again to have both, a pulsed and a modulated CW THz source. Possibly, the method [Filippini] which could rely on the conventional mobile network for transmitting the localisation of the subscriber or the base station would be more efficient. The present single-shot approach should be put in perspective and compared with the methods [Filippini] and [11] as mentioned at the end of Section A.

As noted above, we have inserted the Filippini reference. However, we point out that this reference describes an out-of-band technique, which ours is not. In fact, there are quite a few examples in the literature of out-of-band or legacy band methods. These are, in most cases, quite distinct from what we are discussing. They still require multiple rounds of transmission, not just a single pulse. The achievable localization accuracy with legacy bands is not as good, and may not be sufficient for alignment of narrow pencil-like THz beams (this is a complicated question which probably requires an entire article to address). Most importantly, they do not provide any means for tracking client rotation (i.e., they cannot harvest *both* AoA and AoD information). So the comparison suggested by the referee is not really so meaningful. As for the question of energy efficiency, it's very premature to discuss that in our case, since, as noted elsewhere in this

document, our source of THz radiation is not the one that will be used in any functional system. The power requirements for our femtosecond-laser-based source are irrelevant to any eventual implementation of our protocol in a system.

As an aside, we also note that the reference mentioned by the referee has a very different goal in mind from what we consider. That paper is about initial access. The initial access problem is fundamentally different from our problem statement. In initial access, you want to find any beam configuration so that the access point and client can exchange control signals, mostly for synchronization purposes. The authors of that work assume a split architecture where legacy low-frequency macro cells co-exist with mmWave small cells. In this case, they assume an approximate location of the mobile terminal, estimated with legacy access points. Based on the estimated location, the AP and mobile clients choose a beamwidth level and measure the signal under the geometry-based beam and their neighbor beams. But in our case, we are looking for the best beam configuration that can provide the highest data rate for a point-to-point link. Also we are assuming much less about the cell configuration, in an attempt to be as general as possible. So there is really no way to make a fair comparison between Filippini et al. and our work.

- In this context, I strongly contradict the anticipation of the authors that future THz systems will rely on amplitude or intensity modulation only. Good arguments should be given for such an apodictic statement. I feel that it would be either a waste in spectral resources, or it would require electronic circuits with significantly larger bandwidths. It would be the same idea as the past notion that with the advent of optical amplifiers optical intensity modulation would suffice for the next umpty years. The tendency today is to use higher-order modulation formats like 4QAM or 16QAM. In mobile communications, 4G LTE uses QAM with OFDM, and for 5G this is a still contender.

Despite the referee's assertion, *we did not say* that “future THz systems will rely on amplitude or intensity modulation only.” What we said was that detecting amplitude only, and not phase, “dramatically simplifies the THz node architecture, eliminating the need to keep tight synchronization between the transmitter and receiver, and is robust to small-scale channel variation.” And we also said that “future system designers will prefer to rely on approaches which do not require the detection of phase.”

Of course it may be the case that future systems will use QAM or other modulation schemes which rely on phase detection – we never said otherwise.

However, it is obvious that a sub-system which does not require phase information is much easier to build. For link discovery or other control-plane functions, it is clear that system designers will be happy if they don't need to worry about phase, whether or not the data communication requires phase-sensitive detection. Therefore, it is advantageous to design control-plane protocols which do not rely on phase information. We therefore stand by the statements made in our manuscript.

D. Further comments in sequential order

- What is the authors' definition of a sinc? There are two definitions in use, $\text{sinc}(x) = \sin(x)/x$ or

$$\text{sinc}(x) = \sin(\pi x)/(\pi x)$$

In our understanding, $\text{sinc}(x) = \sin(x)/x$. We were unaware of a second definition. **We have added a notation in the text to remove the ambiguity.**

- The abbreviation PPWG is defined, the abbreviation LWG is not (besides in a caption), and both are used interchangeably.

We have added a definition of the abbreviation LWG, and eliminated all use of PPWG.

- *A "highly directional network" and a 5° direction error at a distance of 15 cm (a G_i of only 27 dB corresponds to a beam angle of 5°) seems to be in contradiction.*

A 5° direction error indicates the uncertainty, due to signal-to-noise limitations, in our estimation of the direction of the client. This has nothing to do with the directionality of the wide-angle broadband broadcast, which, in this case, is not 'highly directional' at all (as shown in figure 1a). Our statements about 'high directionality' obviously do not refer to the same situation as that shown in figure 1a. The referee is confusing control-plane functions (like link discovery, which by its very nature *cannot possibly* be highly directional) with data transmission functions (which will, *of necessity*, be highly directional). Therefore, the contradiction noted by the referee does not exist.

- *Fig. 2 is called a "false color plot". A standard definition is: "A false-color image is an image that depicts an object in colors that differ from those a photograph (a true-color image) would show." If in a grayscale SEM different structures (silicon, metal etc.) are coloured differently, then it is indeed a false colour image. What the authors have done is a colour-coded 3D plot, where the spectral intensity (unfortunately not indicated at the colour-coding scale) is plotted as a function of frequency and angle (again it would have been nice to see which angle is meant - this could have been indicated in Fig. 1b).*

The referee is correct, we should not have used the words "false color plot" in the caption of figure 2. **This has been corrected.** However, the referee's comment about "which angle is meant" is puzzling, since the angle is clearly shown as the horizontal axes in both Figure 2a and 2b. The color coding indicates the amplitude of the measured signal in arbitrary units normalized to unity, so the color-coding scale is irrelevant.

- *In the caption of Fig. 2 it is mentioned that "each row of this false-color image has been normalized to unity magnitude" - I do not understand. Even if I benevolently read "the maximum intensity for each line of constant frequency has been normalized to one" I do not see the justification: The intensity for larger angles will certainly decrease, what harm would it do? And if the scale would suffer, then a logarithmic intensity plot could help.*

The normalization procedure is just as the referee has 'benevolently' stated: the maximum intensity for each line of constant frequency has been normalized to one. We do not see any other way to interpret the statement, and since the referee did indeed interpret it as we intended, it clearly doesn't need to be changed.

As to the question of why we present the data in this way, the answer is already contained in the figure caption – it emphasizes the results at higher frequency which would otherwise be obscured by the frequency-dependence of the input spectrum. It is true that the signal decreases with increasing frequency, due to the nature of our broadband source. But we are looking for changes on top of this decreasing spectrum. So plotting it on a log scale is not nearly as clear, because, while the low-amplitude parts of the signal (at higher frequency) are enhanced by using a log scale, the contrast among those parts is simultaneously washed away. We know this is true because we tried it, and determined that the result is much clearer if presented in the way that we presented it. We're not sure why the referee objects to this – there is no reason not to normalize data as long as one is honest and clear about how it's done (which we were).

- *The "conventional terahertz time-domain spectrometer" should be specified. Is it a Toptica instrument? Which model?*

It is a Picometrix T-ray 4000. However, we don't really feel that it is appropriate for us to advertise a specific manufacturer's product in a journal publication. This is particularly true in this case, since the distinctions among the different manufacturers' products are so minor. All of the manufacturers of these TDS systems (there are currently about 6, worldwide) make about the same instrument with about the same performance specs. So it really doesn't matter which one is used – they really are all about the same, except for details. Thus, we'd rather not be too specific here. However, we're flexible on this point, and would include the name of the specific manufacturer based on the editor's opinion.

- *A "scanning delay line" is mentioned for measuring the ACF for this Fourier spectrometer. How long does the scan take, and what is the electronic processing time? How would it be done for the "rapid link discovery"?*

The answer to this question is the same as the answers to several of the questions above: since we do not expect that future system designers will be putting time-domain spectrometers into wireless access points, the performance specs of the TDS system (i.e. how long does a scan take) are irrelevant to our claims concerning low latency and rapid link discovery.

We note, as an aside, that our spectrometer does not measure an ACF. The operation of terahertz time-domain spectrometers produces a signal which is a convolution of the THz electric field, $E(t)$, with a detector response function. This is not the same as an autocorrelation of $E(t)$, although it does rely on a delay line (as do Fourier transform spectrometers).

Summary:

In its present state, I cannot recommend publication of the submission. However, the approach is interesting enough that it could be worth reading for a wider audience - if the missing data are supplied, if the approach is put properly in perspective, and if all the claims are substantiated, according to the recommendations listed in points A. to D. I recommend a major revision. If space becomes a problem: The (admittedly nice) colourful Fig. 1a could be replaced by a more meaningful schematic setup than is presently displayed in panel Fig. 1b.

We trust that our comments here have laid to rest all of the referee's concerns about missing data and perspective. The main issue is to recognize that we are not claiming to have built a system – that would be dramatically premature. Rather, we are claiming to have demonstrated the feasibility of a completely new protocol. Our results definitively demonstrate that it works, and also show that it could be fast (as opposed to approaches which require sequential sector sweeps, which could not be). The focus of the manuscript is on the demonstration of feasibility, not on the details of how a future system would be built and with what link margin. It would make no sense for us to evaluate things like link margins when we are using a source which is very different from what will eventually be used. More importantly, that evaluation would distract from the main messages of our manuscript, which are (a) that our new protocol actually works, and (b) that there is no obvious impediment to having it incorporated into a future system which requires low latency. We hope that our modifications to the text have clarified our intent sufficiently.

REVIEWERS' COMMENTS:

Reviewer #1 (Remarks to the Author):

I am fine with most clarifications the authors have made, but I would be careful to assert "their impact on systems can be, to first order, neglected". A counterexample is Fig. 5 in [1]: the strongest NLoS path is only 10 dB weaker than the LoS path. It is hard to neglect all NLoS paths. But probably it does not matter since it does not appear in the paper itself.

Authors wrote, "This general approach also has the potential to be able to identify non-line-of-sight paths in a sparse scattering environment, as specular reflections could produce signals with different spectral signatures." Do you mean you expect the approach would fail if the scattering environment is not sparse due to the limited resolution in frequency or do you think it is possible to improve the resolution with the eigenvalue based algorithms (or do you consider this also as out of scope)?

[1] Peng, Bile, Sebastian Rey, and Thomas Kürner. "Channel characteristics study for future indoor millimeter and submillimeter wireless communications." 2016 10th European Conference on Antennas and Propagation (EuCAP). IEEE, 2016.

Reviewer #2 (Remarks to the Author):

A. Instrumentation data

I refer to a comment in Nature 554 (2018) 417-419 by P. Gertler, S. Galiani, and M. Romero, entitled "How to make replication the norm". The authors feel that "the publishing system builds in resistance to replication". While they focus on the fields of economics and life science, their emphasis on the importance of a full documentation of raw data and evaluation code applies also to physical science, and should in this case be supplemented by a documentation of the most important instruments and their relevant specifications (not necessarily the make!), for example as part of the supplementary materials.

B. From scale-model to application

It would be interesting for the reader to see what the problems are when going from a "scale-model" to a practical application. Clearly, the details of this way are not yet known, but such an attempt would give at least an idea of the difficulties to transform a nice idea into a working system. According to my opinion the "feasibility of a new protocol", i.e., the quality of being doable, does not prove its applicability, i.e., its relevance by virtue of being applicable to the matter at hand.

C. Reference [11] just listed thrice

This meant to say that the authors referred to [11] three times, for instance as "an alternative strategy", without giving the reader the faintest idea what this strategy would be. Such a mere listing of references is not really helpful. This has been amended now.

D. Angle

Would it be too much asked to explain what is meant with "angle" on the horizontal axes of the panels in Figure 2? The caption says "emission angle", but it would be much nicer if the authors would point to a proper variable in the setup Fig.1(b).

E. Trivia

I note in passing that in the supplementary materials, page 2, second paragraph, 5th line, there is an inappropriate full stop, "...the TX LWG. located at...". Maybe a comma was meant.

In the last paragraph of the main text, I understand the sentence "...we emphasize that our method relies only on knowledge of the spectrum of the measured signals, and not their spectral phase." as "...we emphasize that our method relies only on knowledge of the amplitude spectrum of the measured signals, and not their spectral phase." Also "power spectrum" instead of "amplitude spectrum" would be okay, but the "spectrum" as such is usually complex.

F. Truth and opinions

A peer reviewer cannot claim to know the absolute truth, and opinions on topics like the ones mentioned above may differ. In this sense I do not want to open a discussion on details, which could cast a shadow on a bright idea. However, I appreciate that the authors spent a few lines on the topic of spectral transmitter power and atmospheric attenuation, on literature issues and perspectives, on techniques to provide power and broadband signals, and on the scope of their scale model.

Summary:

I think that the most important problems have been addressed, and the paper can be published.

2nd round: Response to referee comments

Reviewer #1 (Remarks to the Author):

I am fine with most clarifications the authors have made, but I would be careful to assert “their impact on systems can be, to first order, neglected”. A counterexample is Fig. 5 in [1]: the strongest NLoS path is only 10 dB weaker than the LoS path. It is hard to neglect all NLoS paths. But probably it does not matter since it does not appear in the paper itself.

[1] Peng, Bile, Sebastian Rey, and Thomas Kürner. "Channel characteristics study for future indoor millimeter and submillimeter wireless communications." 2016 10th European Conference on Antennas and Propagation (EuCAP). IEEE, 2016.

We agree that this is a relatively minor issue, since the topic of NLoS paths is not addressed in our manuscript, aside from the comment in the conclusion paragraph quoted below. Yet, we note that the reviewer’s comment actually supports our statement. The ‘counterexample’ cited in the supplied reference shows that the strongest NLoS path is 10 dB weaker than the LoS path. This is not a counterexample, but instead is a *supporting* example. 10 dB is a lot, given that a typical link budget for an indoor LAN might have only 20-30 dB of margin. This means that, *to first order*, these effects can be neglected. This doesn’t mean that it’s an unimportant topic. Indeed, NLoS paths are very important to consider, both for their capacity to detrimentally interfere with the desired transmission and for their possible value in maintaining a link during transient blockage events that temporarily disrupt the LoS path. However, these are all topics that are well beyond the scope of what we are considering in this manuscript. They are not unimportant, just not the very first thing to consider. We are already investigating the impact of NLoS paths in ongoing experiments, with an eye to a subsequent manuscript.

Authors wrote, “This general approach also has the potential to be able to identify non-line-of-sight paths in a sparse scattering environment, as specular reflections could produce signals with different spectral signatures.” Do you mean you expect the approach would fail if the scattering environment is not sparse due to the limited resolution in frequency or do you think it is possible to improve the resolution with the eigenvalue based algorithms (or do you consider this also as out of scope)?

This is a good question. What we mean by that sentence is this: if the scattering environment is not sparse, then a multitude of NLoS paths would produce a multitude of signals, each at a different frequency (since each one arrives at the receiver from a different angle). This would produce a congested spectrum which could be challenging to disentangle. This complex problem is completely different from the conventional approach in the case of rich scattering, since in our case the presence of rich scattering necessarily results in spectral broadening of the received signal (which is not the case in traditional wireless systems, where there is no coupling between the angle of arrival and the frequency). This is a tremendously complex issue, which we regard as FAR beyond the scope of what we have discussed in our manuscript – it is even more challenging than the question of how to handle NLoS paths when they *are* sparse (which, as noted above, we are only just beginning to tackle).

Having said this, we recognize that the sentence quoted by the referee could be modified to offer

some additional clarity on the impact of NLoS effects. Consequently, **we have reworded that sentence, adding a bit of extra discussion.** We do want to avoid opening a whole new can of worms here, which would distract from the main point of our work. But a little extra discussion can't hurt.

Reviewer #2 (Remarks to the Author):

A. Instrumentation data

I refer to a comment in Nature 554 (2018) 417-419 by P. Gertler, S. Galiani, and M. Romero, entitled "How to make replication the norm". The authors feel that "the publishing system builds in resistance to replication". While they focus on the fields of economics and life science, their emphasis on the importance of a full documentation of raw data and evaluation code applies also to physical science, and should in this case be supplemented by a documentation of the most important instruments and their relevant specifications (not necessarily the make!), for example as part of the supplementary materials.

The relevant specifications were already included in the manuscript, in the first part of the Supplementary Materials section (which has now been moved to the main body of the text in a section entitled "Methods"). We believe that nothing is missing here: the text already specifies the integrated output power, the power per bandwidth, the bandwidth, and the spectral resolution employed during these measurements. The input beam's spatial profile (Gaussian) and polarization (TE) were also already specified in the text. For the purpose of replicating these measurements, no other information would be required by any practitioner skilled in the art. There is no "built-in resistance to replication" if all the specs are already there, and only the name of the manufacturer is missing.

B. From scale-model to application

It would be interesting for the reader to see what the problems are when going from a "scale-model" to a practical application. Clearly, the details of this way are not yet known, but such an attempt would give at least an idea of the difficulties to transform a nice idea into a working system. According to my opinion the "feasibility of a new protocol", i.e., the quality of being doable, does not prove its applicability, i.e., its relevance by virtue of being applicable to the matter at hand.

The main step (at least, the first step) in going from a scale model to a practical implementation would be to scale up the power of the transmitter. As noted in our first reply (and in the text that was previously added to the Supplementary Materials document, now the "Methods" section), our transmitter produces something around -135 dBm/Hz of output power. By comparison, existing wireless routers typically emit 100 mW (ignoring EIRP) into a 22 MHz bandwidth, which corresponds to about -53 dBm/Hz. That's more than 80 dB higher. A practical system would need to make up a significant fraction of this power deficit (which seems realistic with current or near-future CMOS technology, fortunately). We feel that the comment added to the Supplementary Materials ("a real system implementation would need to employ transmitters that operate over a narrower (but still very broad) spectral range, but with higher output power, and which are more amenable to integration in a compact package") adequately addresses this

question, at least as the first steps. As the reviewer correctly notes, the details are very much unknown.

We also agree with the reviewer that feasibility does not prove applicability. That's certainly true. However, it is also true that feasibility must be demonstrated first, otherwise nobody will bother to even consider applicability. Proving feasibility and proving applicability are two very distinct tasks. One cannot do both in a single manuscript. We have taken a big step towards the first of those tasks; we hope to follow up with subsequent work that begins to tackle the second.

C. Reference [11] just listed thrice

This meant to say that the authors referred to [11] three times, for instance as "an alternative strategy", without giving the reader the faintest idea what this strategy would be. Such a mere listing of references is not really helpful. This has been amended now.

Agreed.

D. Angle

Would it be too much asked to explain what is meant with "angle" on the horizontal axes of the panels in Figure 2? The caption says "emission angle", but it would be much nicer if the authors would point to a proper variable in the setup Fig.1(b).

The emission angle ϕ_0 is defined in figure 1(b). For clarity, we have modified the caption to Figure 2, specifically mentioning this angle and its definition in Fig 1b.

E. Trivia

I note in passing that in the supplementary materials, page 2, second paragraph, 5th line, there is an inappropriate full stop, "...the TX LWG. located at...". Maybe a comma was meant.

Yes, that was a typo. We fixed it.

In the last paragraph of the main text, I understand the sentence "...we emphasize that our method relies only on knowledge of the spectrum of the measured signals, and not their spectral phase." as "...we emphasize that our method relies only on knowledge of the amplitude spectrum of the measured signals, and not their spectral phase." Also "power spectrum" instead of "amplitude spectrum" would be okay, but the "spectrum" as such is usually complex.

Good point. We changed this to "power spectrum".

F. Truth and opinions

A peer reviewer cannot claim to know the absolute truth, and opinions on topics like the ones mentioned above may differ. In this sense I do not want to open a discussion on details, which could cast a shadow on a bright idea. However, I appreciate that the authors spent a few lines on the topic of spectral transmitter power and atmospheric attenuation, on literature issues and perspectives, on techniques to provide power and broadband signals, and on the scope of their scale model.

Summary:

I think that the most important problems have been addressed, and the paper can be published.

We appreciate the reviewer's efforts, and agree that the resulting modifications have improved the clarity of the manuscript in important ways.